# Inactivation of Multi-Drug Resistant Non-Typhoidal *Salmonella* and Wild-Type *Escherichia coli* STEC Using Organic Acids: A Potential Alternative to the Food Industry

**DOI:** 10.3390/pathogens9100849

**Published:** 2020-10-16

**Authors:** Vinicius Silva Castro, Yhan da Silva Mutz, Denes Kaic Alves Rosario, Adelino Cunha-Neto, Eduardo Eustáquio de Souza Figueiredo, Carlos Adam Conte-Junior

**Affiliations:** 1Instituto de Química, Universidade Federal do Rio de Janeiro, Rio de Janeiro 21941-909, Brazil; viniciuscastro@ufrj.br (V.S.C.); yhan.mutz@ufrj.br (Y.d.S.M.); deneskarosario@ufrj.br (D.K.A.R.); 2Faculdade de Agronomia e Zootecnia, Universidade Federal de Mato Grosso, Mato Grosso 78060-900, Brazil; figueiredoeduardo@ufmt.br; 3Faculdade de Nutrição, Universidade Federal de Mato Grosso, Mato Grosso 78060-900, Brazil; Adenetonut@ufmt.br; 4Departamento de Tecnologia de Alimentos, Faculdade de Veterinária, Universidade Federal Fluminense, Rio de Janeiro 24230-340, Brazil; 5Instituto Nacional de Controle de Qualidade em Saúde, Fundação Oswaldo Cruz, Rio de Janeiro, Rio de Janeiro 21040-900, Brazil

**Keywords:** acetic acid, lactic acid, quaternary of ammonium, sodium hypochlorite, multi-drug resistance

## Abstract

*Salmonella* and *Escherichia coli* are the main bacterial species involved in food outbreaks worldwide. Recent reports showed that chemical sanitizers commonly used to control these pathogens could induce antibiotic resistance. Therefore, this study aimed to describe the efficiency of chemical sanitizers and organic acids when inactivating wild and clinical strains of *Salmonella* and *E. coli*, targeting a 4-log reduction. To achieve this goal, three methods were applied. (i) Disk-diffusion challenge for organic acids. (ii) Determination of MIC for two acids (acetic and lactic), as well as two sanitizers (quaternary compound and sodium hypochlorite). (iii) The development of inactivation models from the previously defined concentrations. In disk-diffusion, the results indicated that wild strains have higher resistance potential when compared to clinical strains. Regarding the models, quaternary ammonium and lactic acid showed a linear pattern of inactivation, while sodium hypochlorite had a linear pattern with tail dispersion, and acetic acid has Weibull dispersion to *E. coli*. The concentration to 4-log reduction differed from *Salmonella* and *E. coli* in acetic acid and sodium hypochlorite. The use of organic acids is an alternative method for antimicrobial control. Our study indicates the levels of organic acids and sanitizers to be used in the inactivation of emerging foodborne pathogens.

## 1. Introduction

*Salmonella* and *Escherichia coli* present in food are a direct risk to human health [1,2,3]. Studies related to food contamination by these pathogens are frequent in the literature [4,5]. In Brazil, as a consequence of food contamination, groups of Shiga-toxin producing *Escherichia coli* (STEC) and non-typhoidal *Salmonella* have been detected in several foodborne cases [6,7]. Moreover, these bacteria groups have several serogroups with the capacity to develop foodborne diseases and potential outbreaks worldwide [8,9]. The pathogenesis of *E. coli* STEC relates to its capacity of causing hemolytic uremic syndrome (HUS) and bloody diarrhea, while non-typhoidal *Salmonella* can cause gastroenteritis (inflammatory condition of the gastrointestinal tract), vomiting, and non-bloody diarrhea [4,10]. For this reason, studies have been made in recent years to inactivate these bacteria at all stages of production [3,11,12,13].

Nowadays, the main challenges in food safety are resistant strains to antimicrobials, such as antibiotics, and sanitizers [14]. The hypothesis is the case that these strains arise due to evolutionary adaptations throughout a selective pressure according to the exposure of bacteria to sanitizers, horizontal transmission of genes, and occasional incorporation of prophage in bacterial DNA [15]. Moreover, an important point in the selection of sanitizers is that gram-negative bacteria have intrinsically high resistance due to the outer plasmatic membrane that hinders the penetration of large molecules [16]. Besides, some sanitizers have been related to the induction of resistance against antibiotics [17].

Chlorine-based compounds, such as sodium hypochlorite, are the most worldwide used sanitizers to inactivate pathogenic bacteria in food and food processing surfaces [18]. Hypochlorous acid (also called “free chlorine”) is approved by the Food and Drug Administration to be used directly in fruit and meat products for the reduction of microbial load [19]. This substance is only recommended in Brazil for fruit and vegetable [20]. Besides, several studies have investigated the antimicrobial capacity of hypochlorous acid and sodium hypochlorite in the application for food decontamination [21,22]. However, a low efficacy in microbial inactivation of sodium hypochlorite (NaOCl) in food has been reported [23,24]. Moreover, according to the European Commission’s expert group for technical advice on organic production, sodium hypochlorite (NaOCl) and other chlorine-based compounds are not recommended for organic farming systems [25] due to the capacity of carcinogenic product formation, such as trihalomethanes [23,24]. Furthermore, the selection of chlorine-resistant bacteria has been reported because of the chlorination step during processing [26]. Moreover, quaternary ammonium compounds are the other sanitizers commonly used by the food industry. However, this sanitizer has been related to the induction of a multi-drug resistance profile in foodborne strains [27,28]. The resistance mechanism is based on efflux pumps for quaternary ammonium compounds. These mechanisms also affect some antibiotics, such as penicillin, gentamicin, trimethoprim, and kanamycin [29].

For this reason, in the food production systems, the application of different disinfectants is required to decrease the microbial load and mitigate the chances of foodborne outbreaks. A promising alternative to overcome the risk of foodborne presence and excessive use of disinfectants are the organic molecules, known as organic acids. Organic acids are a low-cost, easily applicable option in the food industry [30]. Besides, organic acids are generally recognized as safe (GRAS) [31]. Organic acids comprise several molecules, such as acetic, lactic, and citric acids. Their main mechanisms of action are based on cytoplasmic acidifications, the uncoupling of energy production and regulations, and DNA damage [30,32]. The organic acids owe their capacity of penetrating the cell due to their molecule size and polarity [33].

Besides its antimicrobial effect, organic acids have a low impact on the sensory characteristics of products. They are used in animal production, as a growth promoter and in the food industry, for sanitation of equipment directly involved with food. For instance, the use of organic acids may be applied in the sanitation of milk collectors or knives in beef production [34]. In this context, the present study aimed to describe the efficiency of chemical sanitizers and organic acids in the inactivation of wild and clinical strains of non-typhoidal *Salmonella* and *E. coli* STEC, and compare between lethality of the treatment, targeting a 4-log reduction, instead of the classic D value (only adjustable in linear models).

## 2. Results

### 2.1. Organic Acid in DiskxDiffusion Tests

The results of disk-diffusion tests are described in Figure 1. Overall, the acetic and lactic acid had a higher antimicrobial effect in both pathogen strains, while the citric acid has proved to be less effective for both bacteria (*p* < 0.05). When we analyzed *E. coli* (Figure 1a), all wild-type strains were significantly different from the ATCC strain for both, the lactic and acetic acid. However, for the citric acid, only E113-4 and E26-2 strains were different from the ATCC strain (*p* < 0.05). For *Salmonella*, the lactic and acetic acids also had higher inactivation values when compared to citric acid (Figure 1b). For lactic acid, the only difference between strains was found between S3509 and S6130 (*p* < 0.05). For acetic acid, the ATCC and S3509 strains were more susceptible to inactivation (*p* < 0.05). Finally, for citric acid, the same inactivation (*p* > 0.05) was achieved for all strains. In general, the *Salmonella* group (average of all strains) presented the same inactivation pattern than the *E. coli* group for all tested acids (Figure 1c). When evaluating the results of *E. coli* versus *Salmonella* (Figure 1c), it is possible to identify that *Salmonella* is more sensitive to acetic and lactic acid than *E. coli*. However, in citric acid, the same inactivation values (*p* > 0.05) were obtained for both groups.

For subsequent analysis of inactivation kinetics, we selected the *E. coli* strain E26-2 once it showed a difference from the ATCC in the three studied organic acids. Besides, the *Salmonella* strain S6130 was selected because of its difference from ATCC in acetic acid and higher drug-resistant profile among the other *Salmonella* strains [5]. Citric acid presents the lowest inactivation values between acids in both analyzed groups. For that reason, it was not used in the subsequent inactivation modeling.

### 2.2. Inactivation Modeling Using Organic Acid and Sanitizer Treatments

The inactivation kinetics of *Escherichia coli* and *Salmonella* cells were determined by survival models fitted to the survival data (Table 1).

The behavior of *Salmonella* and *Escherichia coli* inactivation is described in Figure 2. The pattern of inactivation was linear for quaternary ammonium, and the rate of inactivation (*Kmax*) resembles in both strains, affirming that the strains are susceptible to this compound. For sodium hypochlorite, the *Salmonella* inactivation model indicates a linear pattern. However, for the *E. coli* strain, a log-linear decrease with a tail in higher concentrations of the sanitizer was found. The *Kmax* of sodium hypochlorite indicates a higher susceptibility in *E. coli* in the initial contact. However, it is possible to verify a resistance in the last three levels used in *E. coli* strains that evidenced the tail effect (an almost constant survival rate) (Figure 2). Concerning organic acid, all models had a linear dispersion for both pathogens and close to the *Kmax* value, with the exception of *E. coli* in acetic acid, where the best data adjust was for the Weibull model, and the *δ* (delta) parameter (concentration required for first log reduction) was obtained. The parameters of each model are described in Table 1.

Results regarding a 4-log reduction of each bacterial load are described in Figure 3. *Escherichia coli* showed a higher (*p* < 0.05) resistance to acetic acid exposure when compared to *Salmonella* (Figure 3). However, the sodium hypochlorite inactivation of the strains showed a contrary behavior, where *E. coli* presented a higher inactivation than *Salmonella* (*p* < 0.05). The quaternary ammonium and lactic acid did not differ among the strains used (*p* < 0.05).

## 3. Discussion

Quaternary ammonium and sodium hypochlorite are commonly applied in the food industry. In the present study, wild *Salmonella* and *E. coli* strains showed a linear pattern of inactivation when exposed to quaternary ammonium. The obtained results indicate an efficacy on the inactivation of *Salmonella* and *E. coli*. This finding is relevant, indicating that the use of this substance remains a good alternative to the inactivation of microbial pathogens. However, the use of quaternary ammonium in farm facilities destined to livestock is still banned in some countries due to the capacity of induction of multi-drug resistance in bacteria [35]. This fact is highlighted for generic efflux pump mechanisms and cassette collectors of resistance. Both are responsible for resistance in some antimicrobial substances too [36]. For example, Deng et al. [37], evaluated *Salmonella* isolates from foods of animal origin at retail, and the results indicated that the use of disinfectants was related to MDR strains by selective pressure and the mechanisms described above. Moreover, other studies have also evidenced that quaternary ammonium compounds induce antimicrobial resistance [27].

Concerning sodium hypochlorite, the results indicate an inactivation tail for the wild *E. coli* strain. This substance is commonly applied in the facilities and directly in products of vegetal origin. The constant use of this substance and, consequently, resistance due to selective pressure, as well as its volatile characteristic, is an additional challenge to be used in the food industry [38]. Our hypothesis for the inactivation tail is based on the saturation of efficacy of the sodium hypochlorite due to the lower availability of free-chlorine or volatilization of this compound. Furthermore, the wild genetic profiles can have an influence because metabolic transcriptions are generally more active [3].

Another point is that sodium hypochlorite is applied to biofilm control in the food industry, which may trigger resistance to this substance since the biofilm populations are dense and the inner layer can survive the use of sanitizer [39]. Our results show the need to evaluate alternative substances for biofilm control, since a population of resistant bacteria may be responsible for several cases of contamination in food [40,41]. It is essential to notice that *Salmonella* had a linear inactivation pattern, which indicates a susceptibility of this wild strain to sodium hypochlorite. According to the study performed by Köhler et al. [42], an efficient reduction in gram-negative bacteria with sodium hypochlorite was found. However, the presence of organic matter and MDR strains was highlighted as limiters of the sanitizer efficiency. For this reason, we performed the assays in Brain Heart Infusion Broth (BHI) broth to simulate the organic matter in the food industry and the time of exposure based on the general protocol used in the facilities.

Organic acids are regarded as an alternative compound for microbiology control in the food industry due to their secure handling, low cost, and quick action [43]. Lactic acid had a linear inactivation pattern in both strains. This finding may be related to the prohibition of the use of lactic acid in the meat production system in Brazil, where the lack of exposure of wild strains to lactic acid in food processing may have led to bacterial strains without previously developed resistance against the acid. Moreover, our results on *E. coli* inactivation kinetics are similar to other studies, pointing out that lactic acid is more effective than acetic acid, although the use of lactic acid in carcass for decontamination in the USA is permitted [44]. Another advantage of lactic acid compared to acetic acid is its volatile characteristics, where lactic acid is odorless, while acetic acid has a strong characteristic odor. Besides, this characteristic of acetic acid can irritate the skin or eyes.

In contrast to lactic acid, a different behavior was found using acetic acid to inactivate *E. coli* strains. Linear inactivation was verified in *Salmonella*, while for *E. coli*, the model of inactivation fitted was non-linear (Weibull). This behavior may be associated with intrinsic resistance of low pH in *E. coli* [45,46]. Moreover, the study performed by Hamdallah et al. [47] shows that an experimental evolution was performed to estimate the adaptation and growth capacity of *E. coli* at adverse pH. The results indicated that pH is the key to transcriptional regulators for acid resistance, and together with selective pressure, it directs the evolution of the strains towards higher resistance profiles. Besides, Chapman and Ross [48] suggested that *Salmonella* and *E. coli* protect themselves against acetic acid by mechanisms that retard acidification of the bacterial cytoplasm. In accordance, our results indicate that *E. coli* may have triggered this previously mentioned mechanism.

On the other hand, for *Salmonella*, these effects were either not elicited or insufficient. It is important to note that the acetic acid is employed in the practice of nutritional supplementation in animals as a growth promoter tool (usually in poultry reared) [49]. This fact may be associated with the resistance to the wild *E. coli* strain in the present study.

Another factor corroborating the discrepancy in acetic acid is the screening step used in the present study. The wild-type strains showed higher resistance to organic acids when compared to the ATCC strains, and *E. coli* was more resistant than *Salmonella* in lactic and acetic acids (Figure 1). A hypothesis is related to the mechanism of the acid tolerance response (ATR) that remains more active in wild strains subjected to abiotic stresses than in clinical strains stored for long periods [3]. The extensive presence of bacteria resistant to several drugs has been related to Brazilian meat and poultry production [5,50,51]. However, the lack of research associated with the use of sanitizers against bacteria with multi-drug resistance in Brazil and the possible exposure to antimicrobials in the food chain emphasizes the need to use wild strains in studies for the inactivation of pathogens. We evaluated different models of inactivation in wild strains that had virulence genotype and occasional multi-drug resistance. The fitted models for organic acid and sanitizers showed suitable adequacy when explaining the microorganism inactivation kinetics. The statistical parameters of adequacy and adjustment were satisfactory, reinforcing the excellent fit to the data (Table 1).

Moreover, the legislation behind the use of disinfectants is still an essential topic of discussion. The direct use or presence of the quaternary of ammonium in food products is prohibited by the ANVISA (Brazil) [52]. The use of sodium hypochlorite for fruit disinfection is the exception [53]. On the other hand, the direct use of organic acids in the processing of beef is allowed by the European Union and the United States of America to decrease the total count of microorganisms [44,54]. However, the use of organic acids is not permitted in Brazil for beef production [43]. The use of organic acid in the animal production system in Brazil is only approved for chicken processing during the sanitization step and the acetic acid as a growth promoter. In this regard, our study encourages the use of organic acid in the food industry, directly or indirectly, as a sanitization process.

Moreover, the use of organic acids is a potential alternative to overcome bacteria antimicrobial resistance [33,55,56]. Antimicrobial resistance has increased the global incidence of infectious diseases, and thus, organic acids, due to the penetration and disintegrating capacity of the outer membrane of gram-negative cells, represent a potential tool to combat this issue [33,55,56]. Besides, it is worth pointing out that the organic acids can be directly applied in food or on food processing surfaces once they are safe for human intake and do not have a daily limit established.

## 4. Materials and Methods

### 4.1. Sample Collection and Preparation

A total of 12 bacterial strains (six *E. coli* and six *Salmonella*) were used and are described in Table 2. The wild-type non-typhoidal *Salmonella* strains used in the study were previously isolated from chicken meat, as described by Cunha-Neto et al. [5], and *E. coli* STEC strains were isolated during the processing of beef by Santos et al. [57]. Besides, the ATTC culture of non-typhoidal *Salmonella* (ATCC-23564) and *Escherichia coli* STEC (ATCC-2196) was used as a reference to address the comparison with wild-type strains. The criteria for inclusion of the strains in the present study were according to the relevance of the serotype (involvement in food outbreaks) and the resistance to one or more classes of antibiotics.

The strains were stored at −80 °C in Brain Heart Infusion Broth (BHI; Kasvi^®^, São Paulo, Brazil), medium with 20% glycerol as stock cultures. Posteriorly, the reactivation of the strains was performed. Briefly, an aliquot of 0.1 mL of the stored culture was collected, inoculated into 9 mL of BHI and incubated at 37 °C for 24 h. Subsequently, a second reactivation round was carried out to maximize the cellular metabolic process.

### 4.2. Selection of Resistant Strains in Organic Acid Using Disk Diffusion Method

To select the strains with the highest resistant profile to be used in the inactivation modeling, a disk diffusion assay for organic acids was performed. Acetic, lactic, and citric acids were standardized for 4096 µg/mL concentrations. Each strain was transferred to Müller-Hinton 2 broth (MH; Himedia^®^, Mumbai, India) and incubated between 2 and 4 h up to 0.5 MacFarland scale [58]. The assay was carried out according to the Kirk Bauer disk-diffusion test [59]. Briefly, the strains were streaked on Müller-Hinton 2 agar (MH; Himedia^®^, India) and diffusion disks (LB; Laborclin^®^, São Paulo, Brazil) with 10 µL of each organic acid were included in the diffusion disks. After the incubation period of 24 h at 37 ± 0.1 °C, the halos were measured.

### 4.3. Organic Acid and Sanitizer Treatments and Enumeration of Survival Cells

According to disk-diffusion results, two strains were selected for the study of the inactivation kinetics *Salmonella* (S45-1) and *Escherichia coli* (E26-2). To determinate the working concentrations, a minimum inhibitory concentration (MIC) was determined for both pathogens. Therefore, MIC was performed by a microdilution test on a 96-well plate. Briefly, 0.2 mL of BHI with *Salmonella* strain at 10^5^ CFU/mL was included in each well. The concentration of the tested organic acids was calculated according to the volume of the total solution. Besides, the doses with a minimum inhibitory concentration for each pathogen were identified, and the doses required for the model were determined. The dose for total inactivation (DTI) was used as the highest working concentration to study the inactivation efficiency of each compound.

Ten concentrations lower than DTI were utilized to study the bacterial inactivation kinetics. The ranges used for acetic and lactic acids were: 4.00–7.50% (*v*/*v*), to sodium hypochlorite: 29.00–70.00% (*v*/*v*) and quaternary ammonium: 0.45–0.68% (*v*/*v*). The analyses were performed following the microdilution method with an exposure time of fifteen-minutes per substance. Briefly, 100 µL of BHI with their respective inoculated bacteria were distributed in 8-wells into a 96-well plate. The percentages of sanitizers and organic acids were calculated according to each point and included in the BHI broth.

After fifteen minutes, the aliquot of 0.1 mL was transferred to 0.9 mL of saline peptone water (for neutralization of the substances). Posterior dilutions were performed, and an aliquot of 0.1 mL was plated on plate count agar (PCA; Kasvi^®^, São Paulo, Brazil). The plates were incubated at 37 °C for 24 h and counted on the electronic plate counter (Eddy-jet-IUL, Barcelona, Spain).

### 4.4. Statistical Analyses and Mathematical Modeling

To evaluate the effects of different organic acids on different strains of *E. coli* and *Salmonella*, as well as to choose the highest organic acid-resistant strains, the data of disk-diffusion assay were analyzed using ANOVA with Tukey’s test. In order to compare *Salmonella* vs. *E. coli* in disk-diffusion inhibition and dose required for 4-log reduction (obtained by inactivation modeling) of each sanitizer and organic acid, a Student’s *t*-test was performed. Models of inactivation were constructed using the software Gina FIT version 1.6 (Katholieke Universiteit Leuven, BEL, Leuven, Belgium). The following models were fitted to the survival data: Log-Linear Bigelow [60], Geeraerd-tail model [61], and the Weibull model [62]. The model evaluation and performance were assessed through the adjusted coefficient of determination (R^2^_adj_) and mean square error (MSE) [63]. The significance level used was 0.05.

## 5. Conclusions

The use of some chemical compounds is being related to the induction of antimicrobial resistance [17], and organic acids are gaining popularity as an alternative strategy for antimicrobial control [33,55,56]. Properties such as low cost, easy handling, fast application, and a non-limited daily intake to consumers pointed to the use of the organic acids. Moreover, organic acids, mainly lactic acid, can be directly used on the beef surface, in the water employed to sanitizations, or in industrial facilities for inactivation of wild strains with resistance to several antimicrobials. Our study indicates the levels of organic acids and sanitizers to be used in the inactivation of emerging foodborne pathogens while using wild-type strains of *E. coli* STEC and *Salmonella,* with a multi-drug resistance profile for the construction of such models that incorporate higher reliability with the expected reduction since they are based on strains with higher resistance profiles.

## Figures and Tables

**Figure 1 pathogens-09-00849-f001:**
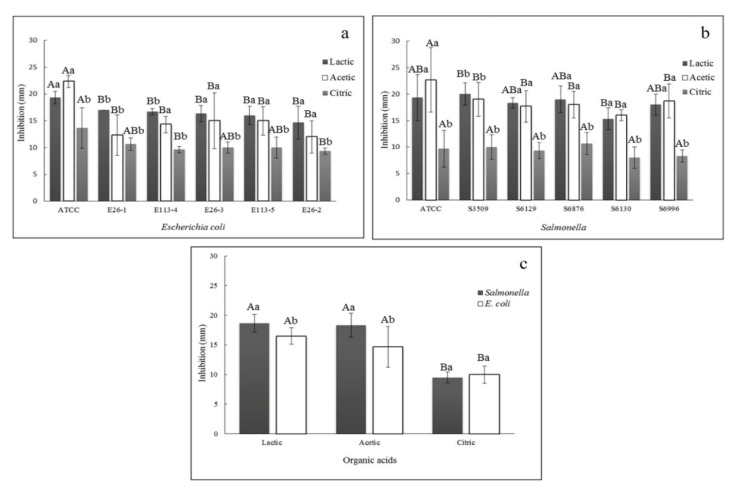
The inhibition zone of *Escherichia coli* and *Salmonella* strains by organic acids. Legend: Average and standard deviation of the *Escherichia coli* STEC (**a**); inhibition and non-typhoidal *Salmonella* strains (**b**)**.** Capital letters indicate the statistical difference (*p* < 0.05) in the same acid in different strains. Lowercase letters indicate the statistical difference (*p* < 0.05) between acids in the same strain. A comparison of *E. coli* and *Salmonella* group (average of all strains) inhibition means (**c**)**.** Capital letters indicate a statistical difference between acids in the same strain, and lowercase letters indicate a statistical difference between strains in the same acid.

**Figure 2 pathogens-09-00849-f002:**
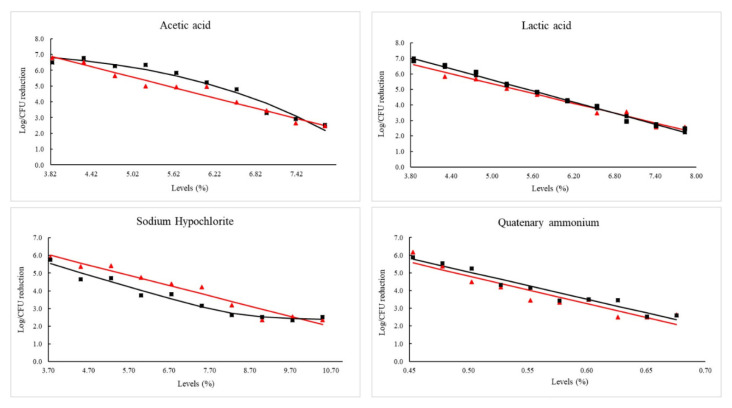
Models fitted to *Salmonella* and *E. coli* inactivation using sanitizers and organic acids. Legend: Experimental data of non-typhoidal *Salmonella* survival is represented by the line and triangle in red, and *E. coli* STEC is represented by the line and squares in black. *Salmonella* used was (S45-1), and *E. coli* was (S26-2). The red line represents the fitted model for *Salmonella* and the black the fitted model for *E. coli*.

**Figure 3 pathogens-09-00849-f003:**
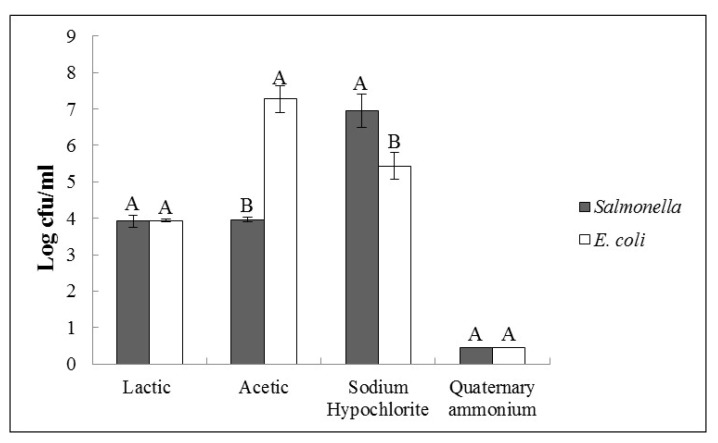
Levels required for each substance reach 4 log-reductions based on the models obtained. Legend: Non-typhoidal *Salmonella* used was S45-1, and *E. coli* STEC used was S26-2. The capital letter indicates a statistical difference in concentration compounds required to 4-log reduction of each substance (column). Adjusted means (LS means) were separated by the Student’s *t*-test. Concentration values are presented as average ± standard deviation.

**Table 1 pathogens-09-00849-t001:** Model parameters and coefficient of determination.

Bacteria	Treatment	Model	4-Log Reduction	R^2^ Adj	*K* _max_	Delta	MSE
*Salmonella*	Lactic acid	Log Linear	3.92	0.97	2.43 ± 0.13	-	0.0511
*Escherichia coli*	Lactic acid	Log Linear	3.92	0.98	2.75 ± 0.11	-	0.0353
*Salmonella*	Acetic acid	Log Linear	3.92	0.96	2.52 ± 0.16	-	0.0803
*Escherichia coli*	Acetic acid	Weibull	7.37	0.95	-	5.03 ± 0.63	0.1162
*Salmonella*	Sodium hypochlorite	Log Linear	6.95	0.94	0.20 ± 0.02	-	0.0971
*Escherichia coli*	Sodium hypochlorite	Log Linear/Tail	5.43	0.96	0.25 ± 0.03	-	0.0524
*Salmonella*	Quaternary ammonium	Log Linear	0.45	0.88	0.54 ± 0.06	-	0.1774
*Escherichia coli*	Quaternary ammonium	Log Linear	0.45	0.94	0.52 ± 0.04	-	0.0765

Legend: *Salmonella* used was (S45-1), and *Escherichia coli* was (S26-2). The model was selected according to the better adjustment of data. 4D reduction: concentration compounds required to 4-log reduction of the bacterial load. R^2^ = adjusted determination coefficient: indicates the goodness of fit. *Kmax*: rate of population inactivation before the tailing effect. Delta: time required for the first log reduction. MSE: mean square error.

**Table 2 pathogens-09-00849-t002:** Bacterial strains used in the present study.

Strain	Resistance Profile ^1^	Isolation Source	Reference
***S.*** **O:4,5 (S45-1)**	AMP, ATM, CFL, CTF, GEN, SUL, SUT, TRI	Chicken meat	[5]
***S.*** **Agona (SAg-2)**	AMP, ATM, CFL, CFO, SUL, SUT, TRI	Chicken meat	[5]
***S.*** **Abony (SAb-3)**	AMP, ATM, CFL, CFO, CTF, SUL, TRI	Chicken meat	[5]
***S.*** **Infantis (SI-4)**	AMP, ATM, CFL, CFO, SUL, TRI	Chicken meat	[5]
***S.*** **Shwarzengrund (SS-5)**	SUL, SUT, TRI	Chicken meat	[5]
***S.*** **Typhimurium (ATCC)**	-	Clinical strain	ATCC–23564
***E. coli*** **O26 (E26-1)**	-	Rectal swab of bovine	[57]
***E. coli*** **O26 (E26-2)**	-	Hide swab	[57]
***E. coli*** **O26 (E26-3)**	-	Rectal swab of bovine	[57]
***E. coli*** **O113:H21 (E113-4)**	STR	Retail beef	[57]
***E. coli*** **O113:H21 (E113-5)**	-	Carcass swab	[57]
***E. coli*** **O26 (ATCC)**	-	Clinical strain	ATCC–2196

Legend: ^1^ AMP, ampicillin; ATM, aztreonam; CFL, cephalothin; CFO, cefoxitin; CTF, ceftiofur (β-lactams); CLO, chloramphenicol; GEN, gentamicin; TET, tetracycline; TRI, trimethoprim; SUL, sulfonamide; STR, streptomycin, and SUT, trimethoprim and sulfamethoxazole (folate pathway inhibitors).

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
