# Peer review of "Inactivation of Multi-Drug Resistant Non-Typhoidal Salmonella and Wild-Type Escherichia coli STEC Using Organic Acids: A Potential Alternative to the Food Industry"

_pathogens, 2020, doi:10.3390/pathogens9100849_

Round 1
Reviewer 1 Report
It would be important to review writing details such as the correct
inclusion of articles such as "the" / "a" etc, throughout the entire
manuscript. Figure 1 presents very complex data, making it difficult to
understand. Perhaps these data could be presented in a table
for better understanding
In the discussion when presenting theories or assumptions
of mechanisms, it would be important to base them with previous
evidence

Author Response
Response to Reviewer 1.
We would like to thank you for your considerations and thoughtful critique of our manuscript. We believe we have responded to all the concerns and suggestions, and thus improved the overall impact of the manuscript.
The modifications are in red font in the text and yellow highlights, and the questions are answered in this file.
R1: It would be important to review writing details such as the correct inclusion of articles such as "the" / "a" etc, throughout the entire manuscript.
AU: Thank you very much for your attention. We reviewed carefully the entire manuscript and corrected the lack of articles in the text and some errors.
R1: Figure 1 presents very complex data, making it difficult to understand. Perhaps these data could be presented in a table for better understanding.
AU: Thank you for your suggestion. When we were writing the manuscript draft, we had chosen to use a table to describe these data; however, as we have three comparisons to be made (Salmonella x Salmonella; E. coli x E. coli and Salmonella x E. coli) the tables were more confused. Therefore, although there may still be a certain level of difficulty in interpreting the results in Figure 1, we believe it has more readability than using the tables to explain these data.
R1: In the discussion when presenting theories or assumptions of mechanisms, it would be important to base them with previous evidence.
AU: Thanks for pointing out this gap. We have included some citations to base our theories and mechanisms listed (see lines: 175, 179, 186, 244, 247, and 319).
Reviewer 2 Report
Dear Editor,
submitted manuscript "Inactivation of multi-drug resistant non-typhoidal Salmonella and wild-type Escherichia coli STEC using organic acids: A potential alternative to the food industry" presents study on the efficiency of chemical sanitizers and organic acids when inactivating wild and clinical strains of Salmonella and E. coli. The current study could be used as applied research for the food industry as well as knowledge to modify existing decontamination procedures.
Some specific comments and remarks to be addressed to Authors:
Introduction and Discussion reveals that Authors are not fully aware of EU legislation vs FDA, FSIS requirements towards decontamination legal requirements, thus may lead to the confusion among readers, particularly,
Page 2, Lines 66-68 statement from references (No. 22 and 24) does not reflect neither EU legislation nor concrete EU member state legal acts, thus some of the statements in the Discussion may also be misleading. Please check original references and the specific content!
Overall, the results are limited to support presented conclusions as well as limitations/ lack of controls may limit acceptance of this manuscript.
Particularly, question arise also on lab tests which were presented on tests on survival of cells: does only plate count was used to check survival cells or any other methods/ controls were used? Alongside, where there any approach used to check for the recovery of stressed cells or used any methods to recover VBNC?
Overall, manuscript could be revised. In addition, through English revision is needed.
Sincerely,
Reviewer #
Author Response
Response to Reviewer 2.
We would like to thank you for your considerations and thoughtful critique of our manuscript. We believe we have responded to all the concerns and suggestions, and thus improved the overall impact of the manuscript.
The modifications are in red font in the text and yellow highlights, and the questions are answered in this file.
R2: Introduction and Discussion reveals that Authors are not fully aware of EU legislation vs FDA, FSIS requirements towards decontamination legal requirements, thus may lead to the confusion among readers, particularly.
AU: Thank you very much for your observation. We rechecked the information and corrected the inconsistencies (see lines: 60-65, 66-69, and 234-237). Also, references have been corrected or included in the references section (see references: 19, 20, 25, 44, and 54).
R2: Page 2, Lines 66-68 statement from references (No. 22 and 24) does not reflect neither EU legislation nor concrete EU member state legal acts, thus some of the statements in the Discussion may also be misleading. Please check original references and the specific content!
AU: Thank you very much for your observation. We corrected the text and included the reference to the Food and Drugs Administration (see line 60, reference 19) and European Commission group (see line 66, reference 25).
R2: Overall, the results are limited to support presented conclusions as well as limitations/ lack of controls may limit acceptance of this manuscript. Particularly, question arise also on lab tests which were presented on tests on survival of cells: does only plate count was used to check survival cells or any other methods/ controls were used? Alongside, where there any approach used to check for the recovery of stressed cells or used any methods to recover VBNC?
AU: Thank you for your feedback; as a way to measure the strain barrier, we use the plate count technique containing non-selective agar (decrease the stress-strain barrier related to selective media). However, these strains are part of a set of studies performed by our research group in which they contain microbiological and molecular information, as described in Table 2.
Regarding VBNC recovery, unfortunately, we did not apply any methodology because we believe it is more related to treatments that use temperature. For this reason, in this study, we made only suggestions in the discussion and did not insert the VBNC statements in the conclusion. In future research, we intend to study the effects of different microbiological control treatments and the induction of VBNC in resistant strains. Regarding the conclusion, the suggested hypotheses were based on the literature, and we included references to support the reader (references: 17, 33, 55, and 56).
R2: Overall, manuscript could be revised. In addition, through English revision is needed.
AU: Thank you very much for your observation. The whole manuscript was revised by a native English speaker.
Round 2
Reviewer 2 Report
Revision and current context of the manuscript is acceptable now.
Reviewer